# A Mitochondrial Genome Phylogeny of Cleridae (Coleoptera, Cleroidea)

**DOI:** 10.3390/insects13020118

**Published:** 2022-01-24

**Authors:** Lilan Yuan, Haoyu Liu, Xueying Ge, Ganyan Yang, Guanglin Xie, Yuxia Yang

**Affiliations:** 1The Key Laboratory of Zoological Systematics and Application, School of Life Science, Institute of Life Science and Green Development, Hebei University, Baoding 071002, China; 201972421@yangtzeu.edu.cn (L.Y.); gexueying@stumail.hbu.edu.cn (X.G.); 2College of Agriculture, Yangtze University, Jingzhou 434025, China; xieguanglin@yangtzeu.edu.cn; 3Beijing Dabu Biotechnology Service Co., Ltd., Beijing 100085, China; yang_ganyan@126.com

**Keywords:** mitochondrial genome, phylogeny, divergence-time estimation, genome biology, Cleridae

## Abstract

**Simple Summary:**

The family Cleridae is a cosmopolitan group with approximately 4000 species and 320 genera. Within the family, the phylogenetic relationships among the subfamilies, and the timing of divergence, remain not yet fully resolved. Mitochondrial genomes have been widely used to reconstruct phylogenies of various insect groups, but never introduced to Cleridae until now. In this study, we newly generated 18 complete or nearly complete mitochondrial genomes, which are conserved in the organization and structure, as well as exhibit typical high A+T-bias and a preference of nucleotides A and G over T and C, as other insects. Further based on these sequences, a phylogeny of this family is reconstructed of different datasets by both maximum likelihood (ML) and Bayesian inference (BI) methods. The results are congruent and support the monophylies of the family and each subfamily, and the subfamilial relationships are recovered as Korynetinae + (Tillinae + (Clerinae + Hydnocerinae)). Moreover, a molecular clock analysis estimated the divergence time of Korynetinae from others no later than 160.18Mya (95% HPD: 158.18–162.06Mya). The current study presents the first mitochondrial genome-based phylogeny of Cleridae, which provides new evidence in reconstructing the phylogenetic relationships among the subfamilies and understanding the mitochondrial features of this family.

**Abstract:**

The predaceous beetle family Cleridae includes a large and widely distributed rapid radiation, which is vital for the ecosystem. Despite its important role, a number of problems remain to be solved regarding the phylogenetic inter-relationships, the timing of divergence, and the mitochondrial biology. Mitochondrial genomes have been widely used to reconstruct phylogenies of various insect groups, but never introduced to Cleridae until now. Here, we generated 18 mitochondrial genomes to address these issues, which are all novel to the family. In addition to phylogenomic analysis, we have leveraged our new sources to study the mitochondrial biology in terms of nucleotide composition, codon usage and substitutional rate, to understand how these vital cellular components may have contributed to the divergence of the Cleridae. Our results recovered Korynetinae sister to the remaining clerids, and the calde of Clerinae+Hydnocerinae is indicated more related to Tillinae. A time-calibrated phylogeny estimated the earliest divergence time of Cleridae was soon after the origin of the family, not later than 160.18 Mya (95% HPD: 158.18–162.07 Mya) during the mid-Jurassic. This is the first mitochondrial genome-based phylogenetic study of the Cleridae that covers nearly all subfamily members, which provides an alternative evidence for reconstructing the phylogenetic relationships.

## 1. Introduction

Cleridae Latreille, 1802, commonly known as checkered beetles, contains a cosmopolitan group (except for the Antarctic) with approximately 4000 species and 320 genera [1,2,3,4]. Most checkered beetles occur on plants and tree trunks, both larvae and adults are evidently predators on other insects, especially wood-infesting beetles and their larvae [4].

Although the taxonomic work can be traced to Linnaeus [5], the systematics of Cleridae has been in a hot debate in the history. Many scholars have made their contributions to this issue, including Spinola [6,7], Lacordaire [8], Schenkling [9,10], Gahan [11], Chapin [12] and Böving and Craighead [13]. The eight subfamilies system proposed by Crowson [14] has been widely accepted until recently, which is largely based on the classification of the latter two works. Later, Winkler [15,16,17] established another two subfamilies and proposed a system of neutral terms as an interim measure to show relatedness among higher taxa, but none of his concepts were followed. The morphology-based classification of Kolibáč [18,19] reduced the number of the subfamilies into four by employing Transformation Series Analysis method, including Tillinae, Hydnocerinae, Clerinae and Korynetinae, which was followed by Leschen [20]. On the contrary, Opitz [1,21] erected another three subfamilies and recognized a total of 12 subfamilies, but no one agreed with his concept. More recently, a molecular phylogeny based on four gene markers was constructed by Gunter et al. [3], who proposed another subfamily, Epiclininae (separated from Clerinae), and meanwhile suggested Thaneroclerinae be reassigned to Cleridae. However, the latter was retained as a separate family by Gimmel et al. [22] as conducted by Kolibáč [19], on basis of a four-gene based phylogeny. In a most recent work [23], Hydnocerinae was downgraded to be a tribe of Clerinae.

Except the differences in higher classification, the phylogenetic relationships within the Cleridae are relatively incongruent. The subfamilial relationships are recovered as Korynetinae + (Hydnocerinae + Clerinae + Tillinae) [19] or Thaneroclerinae +((Tillinae + (Korynetinae + Epiclininae) +(Hydnocerinae + Clerinae))) on the morphological evidence [23], or (Korynetinae + Epiclininae) + ((Thaneroclerinae + Tillinae) + (Hydnocerinae + Clerinae)) if fossil evidence added [23], or Tillinae + ((Thaneroclerinae + Korynetinae + Epiclininae) + (Hydnocerinae + Clerinae)) [3] or Tillinae + ((Korynetinae + Epiclininae) + (Hydnocerinae + Clerinae)) [24,25] on the molecular data. It was argued that, the morphology-based analyses are weighted towards the characters used as taxonomic discriminators between lineages and do not provide an unbiased assessment of the phylognetic relationships [3]. Although some molecular phylogenies were attempted, they were all reconstructed on basis of short nucleotide fragments of mitochondrial *cox1*, *cytb*, *12S* and 16*S* rRNAs as well as nuclear *18S* and *28S* rRNAs [3,23,25,26,27], which may do not contain enough phylogenetic information.

On a more macro-scale, some groups within Cleridae are also interesting for their contribution to the fossil records, and debate continues over when this group first appeared. Even the timing of the emergence of clerid beetles is still debate, with the oldest unambiguous fossil appeared in the Middle Jurassic. A robust timing of the emergence of clerids via phylogeny would be an independent way to test any of these hypotheses, which has profound consequences for our interpretation of the origin and evolution of Cleridae. A number of previous attempts have been made to combine fossil data with molecular evidence in the Coleoptera and gain reliable estimates for the emergence of major clades in this order [28,29,30,31]. Particularly, a detailed time-scaled phylogeny of the superfamily Cleroidea was inferred by Kolibáč [25] most recently, and the origin and divergence time of Cleridae was dated. However, the divergence-time of the clerid subfamilies needs to be reassessed, due to inconsistent results in different studies [25,28,29,30,31].

The paucity of phylogenomic-scale datasets available to address phylogeny prevents large-scale analysis of the Cleridae. Analysis using complete mitochondrial data provides an alternative means of addressing this issue. Mitochondrial genomes contain a range of useful information for phylogenetic investigations, as well as for understanding the basic biology of the organisms that contain them. Their mix of well-conserved and more variable regions render them perfect for understanding the inter-relationships of organism at a range of scales. Furthermore, the crucial roles of mitochondria in providing energy to the cell often means that changes at the genetic level are influenced by the environments inhabited by their species [32]. This direct reflection of evolutionary pressures mean that mitochondrial genome sequences provide a unique window into the biology of the species.

The mitochondrial genome (mitogenome) has been widely used to construct the phylogeny of a number of insect groups [33,34,35,36,37,38,39,40]; however, no studies were carried out on the Cleridae due to a lack of mitochondrial resources. To date, only one complete mitochondrial genome is available in the GenBank [41]. Alongside raw sequence data for phylogenetic comparison, complete mitochondrial genome also present a range of data for other investigations. The mitochondrial biological features, such as compositional bias, codon usage, and substitutional rate variation in mitochondrial genomes provide critical information for molecular evolution [42,43,44,45]. Moreover, phylogenomic analysis with all the 37 mitochondrial genes included has result in improved nodal confidence as compared with single- or multi-locus phylogenetics [33,43,46] but much remains left to be to be tested using such a dataset.

In the present study, we sequenced mitochondrial genomes from 18 species of Cleridae. We used these novel genomes to investigate both the phylogenetic relationships within Cleridae and general biological features of this group. The results will improve our understanding of the phylogeny, divergence time and mitochondrial biology within Cleridae.

## 2. Materials and Methods

### 2.1. Sampling and DNA Extraction

Adult specimens of 18 clerid species (Table 1) were collected and preserved in 100% ethanol at −20 °C before molecular experiments. The specimens were identified following Yang [47]. Total genomic DNAs were extracted using a DNeasy Blood& Tissue kit (QIAGEN, Beijing, China), according to the manufacturer’s instructions. DNAs were stored at −20 °C for long-term storage and further molecular analyses, each species of which was attached with a voucher number and deposited in the Museum of Hebei University, Baoding, China (MHBU).

### 2.2. DNA Sequencing and Assembly

Whole mitochondrial genome sequencing was performed using an Illumina Novaseq 6000 platform (Illumina, Alameda, CA, USA) with 150 bp paired end reads at BerryGenomics, Beijing, China. The sequence reads were first filtered following Zhou et al. [48] and then the remaining high-quality reads were assembled using IDBA-UD [46] under a 98% similarity threshold and k values of a minimum of 40 and a maximum of 160 bp. The gene *cox1* was amplified by polymerase chain reaction (PCR) using universal primers as “reference sequences” to target mitochondrial scaffolds by IDBA-UD [49] to acquire the best-fit, which is under at least 98% similarity. The PCR cycling conditions comprised a predenaturation at 94 °C for 5 min and 35 cycles of denaturation at 94 °C for 50 s, annealing at 48 °C for 45 s and elongation at 72 °C for 8 min at the end of all cycles. Geneious 2019.2 [50] software was used to manually map the clean readings on the obtained mitochondrial scaffolds to check the accuracy of the assembly.

### 2.3. Sequence Annotation and Analyses

Gene annotation was carried out by Geneious 2019.2 [50] software and the MITOS2 webserver (Available at http://mitos2.bioinf.uni-leipzig.de/index.py (accessed on 1 Feb. 2021) [51]. The circular map of the mitochondrial genome was produced using a visualization tool OrganellarGenomeDRAW (http://ogdraw.mpimp-golm.mpg.de/index.shtml (accessed on 3 April 2021) [52]. Base composition, component skew, codon usage, and relative synonymous codon usage (RSCU) were analyzed by PhyloSuitev1.2.2 [53]. DnaSPv5.10.01 [54] was used to estimate the nucleotide diversity (Pi) in a sliding window analysis (a sliding window of 200 bp and a step size of 20 bp) and non-synonymous (Ka)/synonymous (Ks) substitution rates among the 13 protein-coding genes (PCGs). The genetic distances were computed using MEGA 7.0 [55] with the Kimura-2-parameter model. Substitution saturation of each codon position of PCGs was measured based on Xia’s test [56] implemented in DAMBE program v6.4.81 [57]. SymTest v2.0.47 with Bowker’s matching pair symmetry test [58] was used to analyze the differences of heterogeneous sequences in the datasets, and the heat maps were generated according to the inferred *p*-values. The tandem repeats finder program (http://tandem.bu.edu/trf/trf.html (accessed on 1 April 2021) was used to predict the tandem repeat elements in the A+T-rich region [59].

### 2.4. Phylogenetic Analyses

Mitochondrial genomes of 18 species representing all four subfamilies of Cleridae were selected, which are all novel in this family (Table 1). The one sequence available in the GenBank was not included in the analyses due to possible misidentification [41]. Two species of Dasytinae in Melyridae *sensu*
*lato* were chosen as the outgroups [60,61]. To test the impact of the third codon position of the PCGs and gene combination types on the phylogenetic analysis, four datasets were concatenated: (i) the first and second codon positions of 13 PCGs (PCG12); (ii) all three codon positions of PCGs (PCG); (iii) all PCGs and rRNAs (PCGrRNA); (iv) all PCGs, rRNAs and tRNAs (PCGRNA).

Alignment of PCGs, tRNAs and rRNAs was performed by using Mafftv7.313 [62] in PhyloSuite v1.2.2 (alignment strategy: auto) [53]. Intergeneric gaps and ambiguous sites were removed using Gblocks v 0.91b [63], and individual alignments were concatenated using PhyloSuite. All matrices were analyzed using maximum likelihood (ML) with IQ-TREE v1.6.12 [64] on the dedicated webserver (Available at http://iqtree.cibiv.univie.ac.at/ (accessed on 10 June 2021) and Bayesian inference (BI) with MrBayesv3.2.6 [65] on CIPRES web server (Available at https://www.phylo.org/ (accessed on 25 May 2021) or MrBayesv3.2.6 [65] in PhyloSuite, respectively. A 1000 replicate bootstrapping was performed for ML analyses using the “ultrafast” option [66] implemented in IQ-TREE, with the SH-alerts test used to assess branch supporting values. The best model (Appendix A) was inferred by Partition-Finder (v2.1.1) [67]. Four simultaneous Markov chain Monte Carlo (MCMC) runs of 1 million generations twice, with trees sampled every one thousand generations, and the first 25% of 1 million generations twice, with trees sampled every one thousand generations, and the first 25% of steps were discarded as burn-in. Stationarity was considered to be reached when the average standard deviation of split frequencies was below 0.01. Trees produced from all analyses were visualized and edited using iTOL (https://itol.embl.de (accessed on 1 July 2021) [68].

### 2.5. Divergence Time Estimate

Divergence times among subfamilies were estimated using the nucleotide sequences of 13 PCGs with a relaxed clock log normal model in BEAST1.10.4 [69,70]. We adopted the Calibrated Yule model for the prior tree, and used the GTR+I+G for concatenation by Phylosuite v1.2.2. For estimating divergence time calibration, *Protoclerus*
*korynetoides*, the oldest reported fossil of Cleridae from the Middle Jurassic in NE China (mean value of normal prior distribution c.160.2 Mya, SD = 1.0) [71] was used to assign age calibration. The final Markov chain was run twice for each 1 × 10^8^ generations, sampling every 10,000 generations with the first 25% of generations discarded as burn-in, after confirming the convergence of chains with Tracer v.1.7.2 [72]. The effective sample size of the majority of parameters was >200. We summarized the subsamples trees in a maximum clade credibility tree with mean heights using Tree Annotator v1.10.4, and then the mean heights and 95% highest probability density (95%HPD) were displayed in Figtree v1.4.3 [73].

## 3. Results

### 3.1. Phylogenetic Analyses

Heterogeneity of nucleotide divergence was examined under pairwise comparisons in a multiple sequence alignment. The heterogeneous sequence divergence of PCG12 dataset (Figure 1a) is much lower than that of the other three datasets (Figure 1b–d), indicating that the third codon positions are more rate-heterogeneous than the first and second ones. Furthermore, substitution of the three codon positions of PCGs are generally not saturated, except for the third codon position (Figure 2a–c), suggesting more substitution in the third codon positions than in the first and second codon positions.

Analysis of the four datasets resulted in nearly identical and fully resolved topologies with high nodal support values under both ML and BI analyses (Figure 2 and Appendix A). Compared with all others (Appendix A; BSs = 55–57, PPs = 0.752–0.831), the nodal support values of Hydnocerinae (BS = 72, PP = 1) were improved by both ML and BI analyses of the PCG12 dataset when the third codon positions were excluded (Figure 2d).

In all phylogenetic analyses, the monophyly of the family and each subfamily is well supported (PPs = 1, BSs = 100) (Figure 2 and Appendix A). Korynetinae is recovered next to the remaining clerids with high support value (PPs = 1, BSs = 100). Hydnocerinae and Clerinae are recovered as sister groups, which is greatly supported (PPs = 1, BSs = 100). The clade of Hydnocerinae + Clerinae is sister to Tillinae, which is highly supported (PPs = 1, BSs = 100).

### 3.2. Divergence-Time Estimation

The age estimates (average and 95% PHD) of each subfamily based on the topology recovered from BEAST analysis were summarized in Figure 3. The BEAST analysis indicated that the divergence events of Cleridae occurred during the mid-Jurassic and mid-Cretaceous period.

The subfamily Korynetinae was divergent from all others soon after the origin of the family during the mid-Jurassic, approximately at 160.18 Mya (95% HPD: 158.18–162.07 Mya). After a period of evolution, the Tillinae split from the rest during the early Cretaceous, at 138.58 Mya (95% HPD: 123.00–147.69 Mya), and finally Clerinae and Hydnocerinae became divided at 109.03 Mya (95% HPD: 92.80–123.02 Mya).

### 3.3. General Features of Mitochondrial Genome

The complete or nearly complete mitochondrial genomes of 18 clerid species were successfully sequenced. It is a double-strand circular molecule, which is made up of 37 genes, including 13 PCGs, 22 tRNA genes, 2 rRNA genes, and an A+T-rich region (or control region) (Appendix A), of which 14 genes (8 tRNAs, 4 PCGs and 2 rRNAs) were transcribed on the minority strand (N-strand), whereas the rest (14 tRNAs and 9PCGs) on the majority strand (J-strand).

The complete mitogenomes of clerids range from 15,638 bp to 17,127 bp in size (Appendix A). The sizes of the control region vary greatly among different species (ranging from 987bp to 2,401 bp), whereas the PCGs, tRNAs, and rRNAs show little variation in length (Appendix A).

### 3.4. Nucleotide Composition

In the full genomes, the A+T contents range from 77.1% to 80%, those of the rRNAs range from 81.2% to 83.4%, and those of the control region from 84.5% to 90.3% in Cleridae (Appendix A).

Comparison among the PCGs (Figure 4b, Appendix A) shows that the average A+T content of *cox1* is the lowest (68.5%) in all clerids, followed by *cox3* (72.5%) and *cytb* (73.5%), whereas those of *apt8* and *nad6* are the highest (85.0% and 84.7%, respectively). Additionally, the A+T contents of the third codon position of PCGs, ranging from 86.04% to 93.78%, are much higher than those of the other two codon positions, which range from 68.19% to 72.42% (Figure 4a, Appendix A). Moreover, the nucleotide skew analysis (Figure 4c,d, Appendix A) shows that the AT skews are positive for most species of Cleridae, whereas the GC skews are all negative. The correlations of Cleridae’s mitogenomes were calculated between A+T content versus AT skew (y = −0.0076x + 0.5923, R^2^ = 0.2771), and G+C content versus GC skew (y = −0.0184x + 0.2618, R^2^ = 0.6213), respectively. Both of them showed negative linear correlations, implying that the quantity of A+T becomes more equivalent with the increase in A+T content, whereas G+C show a larger quantity gap with the increase in G+C content.

### 3.5. Codon Usage and Evolutionary Rates

In total, six initiation codons (ATA, ATT, ATG, ATC, TTG, GTG) were used in encoding the PCGs of Cleridae, which are terminated with TAG or TAA codons or truncated T codons (Appendix A). Comparison of these codons among PCGs shows that *atp6* always uses ATA, and *cox3* and *cytb* use ATG as start codons, respectively. For the stop codons, *atp6, atp8*, and *nad6* all use TAA, and nad4 uses T-, respectively. The most frequently used start codons are ATT and ATA, and the stop codon is TAA.

Relative synonymous codon usage (RSCU) shows that all synonymous codons of 22 amino acids are present in Cleridae. Among these codons, UUA-Leu2 and UCU-Ser2 are the first two frequently used codons, followed by CCU-Pro, GCU-Ala, and CGA-Arg (Appendix A). The RSCU values of the PCGs reveal that there is a higher frequency in the usage of AT than that of GC in the third codon positions. In Cleridae (Appendix A), Leu2, Ile, and Phe are the most frequently encoded amino acids (over 10%), followed by Met, Ser2, and Asn.

We conducted pairwise non-synonymous (Ka) to synonymous (Ks) substitution ratio (ω) analyses for Cleridae (Figure 5a), and found that the Ka/Ks of all PCGs are less than 1. Among the PCGs, *cox2* has lowest value (ω = 0.134), whereas those of the *nad* family genes (ω = 0.25 − 0.543) and *atp8* (ω = 0.496) are higher. The pairwise genetic distance calculation (Figure 5a) indicates that *cox2* (0.149), *cytb* (0.15) and *nad4l* (0.152) are the lower values, whereas *atp8* (0.244), *nad2* (0.244) and *nad6* (0.259) are the three highest.

Sliding window analysis was implemented to study the nucleotide diversity of 13 PCGs exhibited in Figure 5b. Among the genes, *nad6* (Pi = 0.26) has the highest variability, followed by *atp8* and *nad2* (both Pi = 0.24); *cox2*, *cytb*, and *nad4l* (all Pi = 0.15) have the lowest variability. This result is roughly congruent with that of the above pairwise genetic distance calculation.

## 4. Discussion

### 4.1. Phylogeny and Divergence-Time Estimation

Comparing all the topologies produced by different datasets by both ML and BI analyses, the nodal support values were improved based on the PCG12 dataset when the third codon positions were excluded. This may be a result of the high heterogeneity (Figure 1) and saturated substitution (Figure 2c) of the third codon positions, which is relatively free in evolution [74], and the change of nucleotide in this position rarely change the amino acid product, consistent with their codon usage (Appendix A). When the third codon positions are highly heterogeneous or saturated, they should be excluded from the phylogeny reconstruction, generally because they will affect the reliability of the phylogenetic analysis results or be less informative, as suggested by other studies [75,76,77,78].

The monophyly of the family and each subfamily is well supported, which is consistent with some phylogenetic studies [2,19,28]. Korynetinae is recovered next to the remaining clerids with high support value, which is congruent with the morphology-based phylogeny carried out by Kolibáč [19]. Korynetinae was defined by a synapomophy (reduction in size of the fourth tarsomeres) by Kolibáč [19]. However, when the fossil evidence was implemented, Thaneroclerinae was recovered as the basal clade of Cleridae [27]. Otherwise, the molecular phylogenies suggested Tillinae were at the base of the clerid clade [3,23,25,28], regardless of whether Thaneroclerinae was sampled or not. Tillinae is the only clerid subfamily in which the procoxal cavities are closed internally, which is a synapomorphy of this subfamily [1] but may be apomorphic within Cleridae, since the Cleroid family Rentoniidae is also equipped with this feature [3].

Hydnocerinae and Clerinae are recovered as sister groups, which is consistent with the results of some studies [2,28], but they were suggested to be paraphyletic by others [3,23,25,78]. Based on the latter results, Hydnocerinae was synonymized with Clerinae by Bartlett [23]. Although our taxa sampling is too limited to test the monophylies of these two subfamilies, Opitz’s [1] findings indicated that Hydnocerinae and Clerinae both possess two secondary stomodaeal valve lobes (four in Tillinae and Thaneroclerinae or completely reduced in Korynetinae), which is synapomorphic for supporting their monophyly or sister relationship.

The clade of Hydnocerinae + Clerinae is sister to Tillinae, which is highly supported and shows some similarity to the phylogeny when the fossil data were implemented [78]. However, Epiclininae, which was separated from Clerinae, together with Korynetinae recovered a sister relationship with this clade [3,25]. In terms of our results, Korynetinae is relatively distant from this clade in the affinity, although no Epiclininae was available for this study.

The estimated divergence time is earlier than those predicted by previous studies [28,29,31], while later than others [25,30]. *Protoclerus*
*korynetoides,* one of the two earliest clerid fossil was not attributed to any subfamily [24] but was used to assign age calibration prior for the family in the present study, so the estimated time is later than in theory at least. Therefore, the origin of Cleridae is probably from the early Jurassic period, as predicted by Toussaint et al. [30] and Kolibáč et al. [25], and accordingly, the divergence time of the subfamilies was also earlier than that estimated in this study. 

### 4.2. The Characteristics of Mitochondrial Genomes

All typical animal mitochondrial genes and control regions were identified in the 18 mitogenomes, and they are arranged in the same order as the hypothetical ancestral insect [79], indicating that the mitogenomes are highly conserved in Cleridae.

The sizes of the whole mitogenome among the species are comparable, in which the length of the control region varies greatly, whereas other components show little variation. This suggests that the mitogenome size of different clerids is largely determined by the size of control regions, such as other insects [80].

As with all other insects [36], the mitochondrial genomes of clerids exhibit the typical high A+T-bias either in the full genome or different components or different positions of PCGs (Figure 2a), with the A+T contents all higher than 68.19% (Appendix A). Additionally, Cleridae usually have a preference of A and G over T and C in the mitogenomes. The causes for such base composition bias are multifactorial, but most of the hypotheses suggest that the asymmetric nucleotide composition is the result of mutations and selection pressures [81], and the value of the GC-skew of the insect mitochondrial genomes seems to be associated with the replication orientation [82].

The ratio (ω) of Ka/Ks can be used to estimate whether a sequence is undergoing purifying, neutral, or positive selection [83]. We found that all PCGs are evolving under a purifying selection (ω < 1), and *cox2* exhibited the strongest purifying selection, whereas the *nad* family genes and *atp8* relaxed. Furthermore, the results of pairwise genetic distance calculation and the nucleotide diversity analysis are consistent, indicating that *cox2*, *cytb*, and *nad4l* evolve comparatively slowly or have lower variability, whereas *atp8*, *nad2*, and *nad6* are evolving faster or have higher variability. Nucleotide diversity analyses are critical for designing species-specific markers useful in taxa where morphological identification is difficult and ambiguous [84,85,86]. Usually, *cox1* is the last variable and can be a potential marker for species identification and has been widely used in the taxonomic work of insects [87,88]. However, our analysis reveals that *cox2* may be more suitable for Cleridae. Given this result, we suggest that the gene markers should be designed for different families or even for different subfamilies in the taxonomy if necessary.

## 5. Conclusions

The previous work attempted to resolve the phylogenetic relationship of Cleridae exclusively based on morphological characters or short nucleotide fragments. With this study, we documented the phylogenetic relationships of Cleridae based on the complete mitochondrial genomes. We chose the Chinese species as a start point, because they were the most accessible for us, with 18 species representing four subfamilies providing a diverse, but not unmanageable, number of taxa for analysis. Nevertheless, the molecular phylogenies, including this study, were analyzed on the basis of a minority part of species or limited molecular data in comparison with an estimated 4000 species of checkered beetles worldwide. Therefore, many more species need to be included in future analysis to establish a solid and dependable classification of Cleridae, especially for some taxa (i.e., Thanerocleri-dae/-nae, Epiclininae) whose status is controversial. In particular, the complete mitochondrial genomes should be encouraged to accumulate more for Cleridae, in view of their high value in investigating phylogenetic relationships of the insects.

## Figures and Tables

**Figure 1 insects-13-00118-f001:**
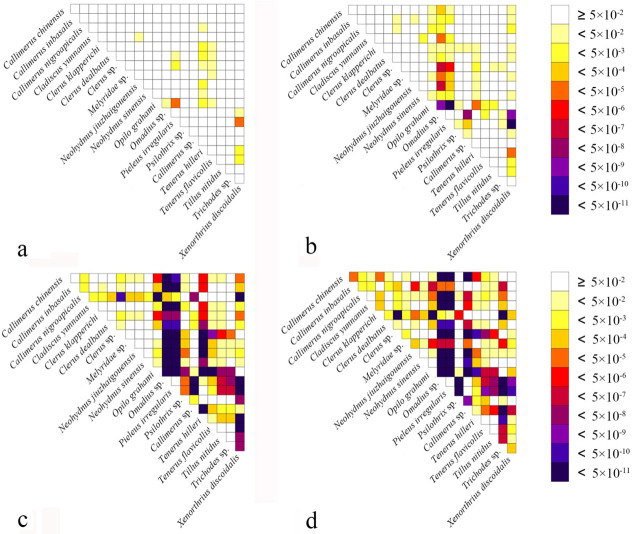
Heterogeneous sequence divergence of mitochondrial genomes of Cleridae resulting from pairwise comparison of four aligned datasets: (**a**) PCG12rRNA; (**b**) PCG; (**c**) PCGrRNA; (**d**) PCGRNA. The dark colors indicate the higher randomized accordance, whereas the lighter colors indicate the opposite. All taxa names are listed to the right of the heat map. Although cells specify *p*-values > 0.05, indicating that corresponding pairs of nucleotide sequences do not violate the assumption of global stationery and homogeneity conditions.

**Figure 2 insects-13-00118-f002:**
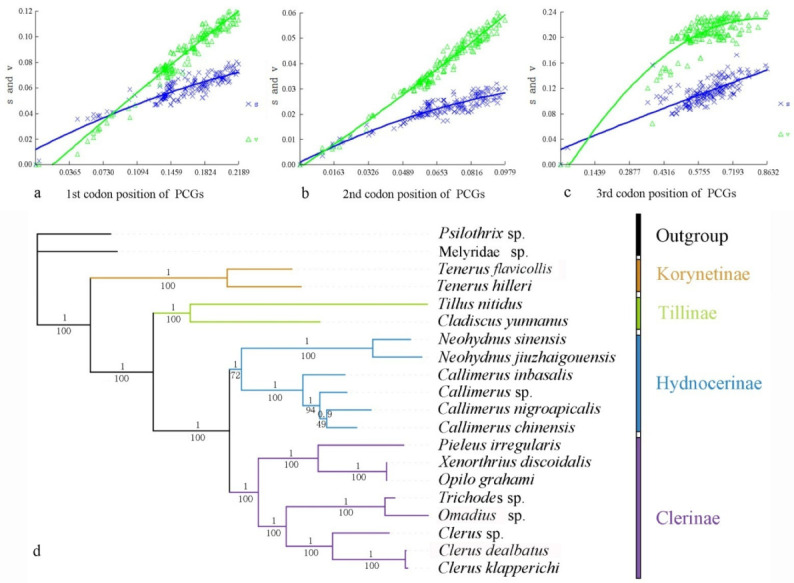
Nucleotide substitution saturation plots of all mitochondrial protein-coding genes: (**a**) 1st codon positions; (**b**) second codon positions; (**c**) third codon positions. Plots in blue and green indicate transition and transversion, respectively; (**d**) phylogenetic tree of Cleridae inferred from the BI and ML analyses of the PCG12RNAdataset. Numbers at the branches determine posterior probability (upper) or bootstrap value (lower).

**Figure 3 insects-13-00118-f003:**
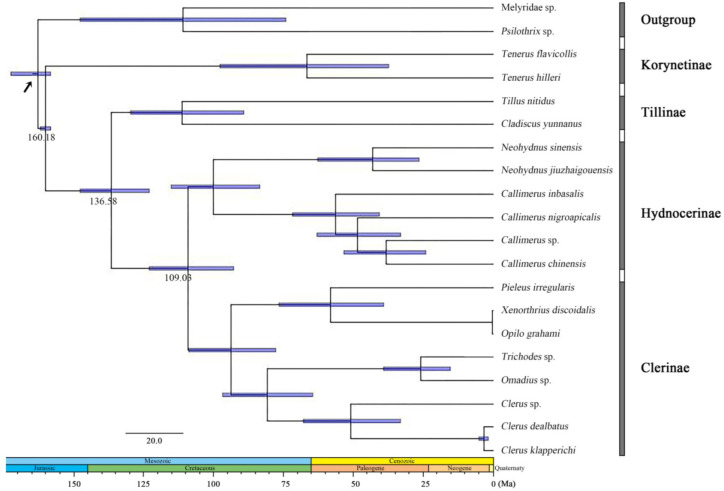
Chronogram with estimated divergence time based on fixed rate calibration amongst Cleridae using BEAST1.10.4. Horizontal bars represent 95% credibility intervals of time estimates. Numbers on the nodes indicate the mean divergence times. Calibration point is marked by an arrow.

**Figure 4 insects-13-00118-f004:**
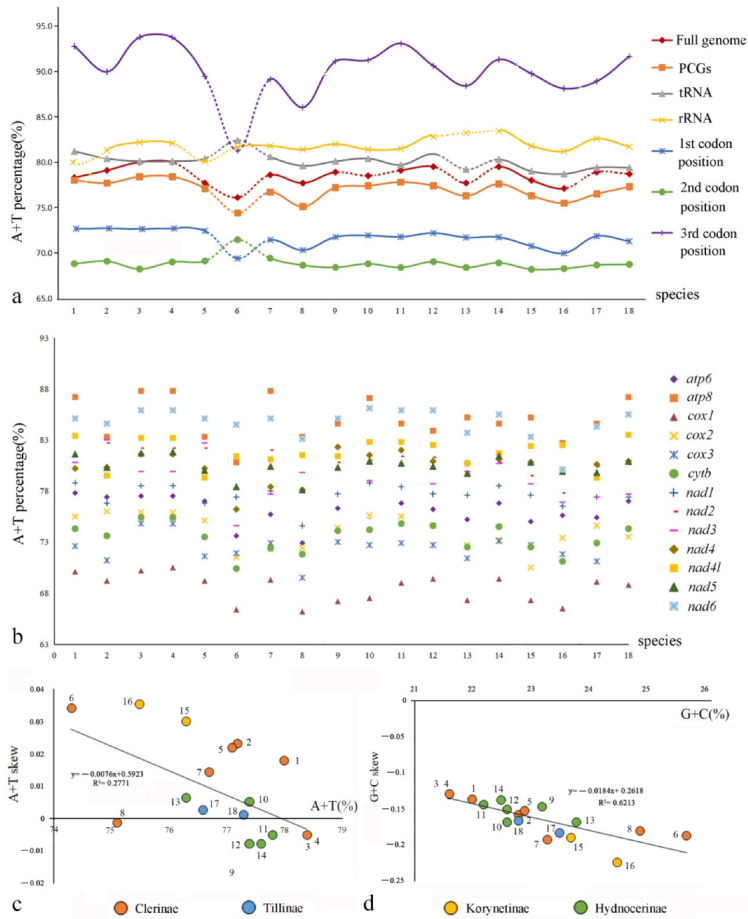
Comparison of nucleotide compositions of mitochondrial genomes among four subfamilies of Cleridae: (**a**) A+T% of different components or positions; (**b**) A+T% of different PCGs; (**c**) the correlations between A+T% and AT skew in the 13 PCGs; (**d**) the correlations between G+C% and GC skew in the 13 PCGs.

**Figure 5 insects-13-00118-f005:**
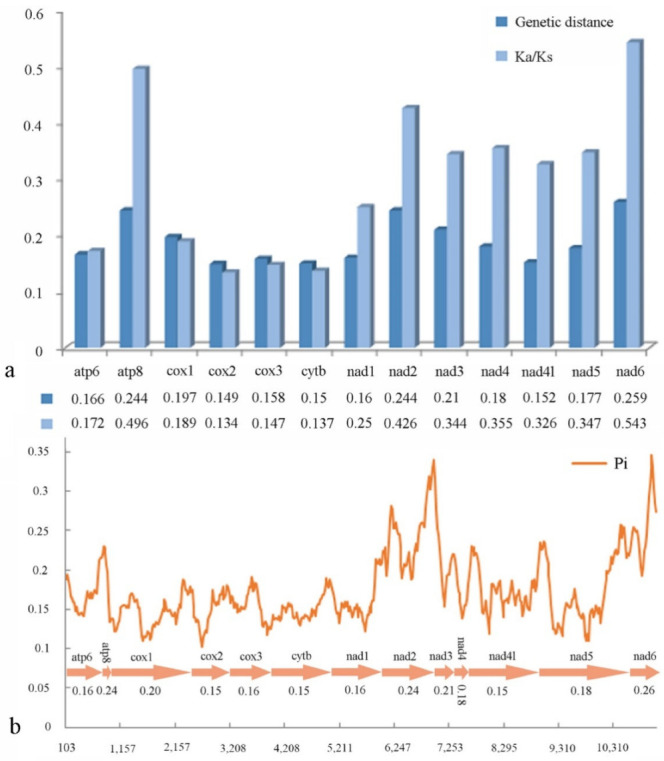
(**a**) Ka/Ks ratios and genetic distances; (**b**) nucleotide diversity of PCGs of Cleridae. The polyline represents the value of nucleotide diversity (a sliding window of 200 bp with the step size of 20 bp), and the average value for each gene is shown under the gene name.

**Table 1 insects-13-00118-t001:** Taxonomic information and GenBank accession numbers of mitochondrial genomes used in the study.

Species ID	Family/ Subfamily	Species	Depository/ Voucher No.	Locality/ Collection Information	Geographic Coordinates	GenBank No.
1	Cleridae/ Clerinae	*Clerus* sp.	MHBU, 2CA0214	China, Beijing: Mentougou, Xiaolongmen, V-12-2018	115°27′4.22″ E, 39°58′36.87″ N	MZ464014
2		*Clerus dealbatus*	MHBU, CAN0129	China, Shaanxi: Yangxian, Youdeng, VI-24-2017	107°22′30.72″ E, 33°27′7.80″ N	MZ490582
3		*Xenorthrius discoidalis*	MHBU, CAN0019	China, Gansu: Tianshui, Maiji, Dangchuan, Fangmatan, VIII-9-2018	106°6′48.37″ E, 34°25′3.38″ N	MZ490583
4		*Opilo grahami*	MHBU, CAN0176	China, Yunnan: Jingdong, Ailaoshan, Xujiaba, VIII-17-2013	101°1′14.74″ E, 24°31′15.90″ N	MZ488575
5		*Clerus klapperichi*	MHBU, 2CA0167	China, Zhejiang: Tianmushan, Xianrending, V-6-2018	119°26′43.64″ E, 30°20′43.87″ N	MZ475053
6		*Omadius* sp.	MHBU, 2CA0218	China, Xizang: Nyingchi, Medôg, VIII-14-2016,	95°20′30.47″ E, 29°19′55.29″ N	MZ490580
7		*Trichodes* sp.	MHBU, CAN0128	China, Shaanxi: Yangxian, Youdengvill., VI-24-2017	107°22′30.95″ E, 33°27′7.72″ N	MZ490584
8		*Pieleus irregularis*	MHBU, 2CA0165	China, Zhejiang: Tianmushan, Xianrending, V-6-2018	119°26′43.64″ E, 30°20′43.87″ N	MZ488576
9	Cleridae/ Hydnocerinae	*Callimerus chinensis*	MHBU, CAN0173	China, Yunnan: Lancang, Donghe, Shangbanggan, XI-18-2017	100°04′06.01″ E, 55°44′09″ N	MZ464016
10		*Callimerus inbasalis*	MHBU, 2CA0173	China, Yunnan: Puer, Lancang, VII-5-2017	99°56′2.49″ E, 22°33′33.22″ N	MZ464017
11		*Callimerus* sp.	MHBU, 2CA0166	China, Hunan: Shaoyang, Chengbu, Dankou, Taiping, V-6-2018	110°14′52.64″ E, 26°21′25.14″ N	MZ488577
12		*Callimerus nigroapicalis*	MHBU, CAN0183	China, Hainan: Ledong, Jiangfengling, IV-10-2019	108°54′32.99″ E, 18°43′49.69″ N	MZ475052
13		*Neohydnus sinensis*	MHBU, 2CA0035	China, Guangxi: Wuming, Damingshan, V-21-2011	108°20′33.57″ E, 23°31′45.78″ N	MZ464019
14		*Neohydnus jiuzhaigouensis*	MHBU, CAN0226	China, Hubei: Shennongjia, Tiechanghe, VI-25-2019	110°46′15.17″ E, 31°39′51.54″ N	MZ464018
15	Cleridae/ Korynetinae	*Tenerus* *flavicollis*	MHBU, 2CA0146	China, Yunnan: IV-29-2010	102°55′40.02″ E, 25°0′3.18″ N	MZ488578
16		*Tenerus hilleri*	MHBU, 2CA0172	China, Sichuan: Pengzhou, Danjingshan, VI-7-2019,	103°50′21.24″ E, 31°5′9.99″ N	MZ488579
17	Cleridae/ Tillinae	*Tillus nitidus*	MHBU, 2CA0216	China, Shaanxi: Zhouzhi, Louguantai, VI-25-2008	108°19′58.28″ E, 34°3′49.19″ N	MZ490581
18		*Cladiscus yunnanus*	MHBU, 2CA0079	China, Yunnan: Xishuangbanna, Tropical Botanical Garden, VI-2-2015	101°16′44.26″ E, 21°55′20.83″ N	MZ464015
Out- group	Melyridae/ Dasytinae	Dasytinae sp. *Psilothrix* sp.				JX412765 JX412801

## Data Availability

The data presented in this study are available in the article and in the Appendix A.

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
