# Peer review of "A Mitochondrial Genome Phylogeny of Cleridae (Coleoptera, Cleroidea)"

_insects, 2022, doi:10.3390/insects13020118_

Round 1

Reviewer 1 Report

The article is very interesting, original and with high scientifical value. I fully recommend it for publishing in Insects.

  1. I am not a native English speaker but I consider some formulation unclear or incorrect. A part of them is highlighted but surely not all. Some misspellings/typos are also highlighted in yellow.
  2. The reason why I marked the manuscript as "major revision" is that the authors did not compare their molecular dating with the paper by Kolibac et al.: Extinct and extant Pacific Trogossitidae and the
    evolution of Cleroidea (Coleoptera) after the Late
    Triassic biotic crisis. Zoological Journal of the Linnean Society, 2021, 191, 846–882. In this paper, the molecular dating of all major cleroid clades is introduced, including that for the all clerid subfamilies; therefore author's attempt for dating is not the first one. The authors probably have not read the recently published article (online Sept. 2020, print March 2021) which should be cited, phylogenies compared (Introduction) and especially discussed in the section 3.2. Divergence-time estimation. Results of the both datings are essentially compatible but should be thoroughly compared.
  3. In 2.5. Divergence time estimate section, P. korynetoides and Trichodes stebingeri are mentioned as the calibration fossils. However, the later is not pictured by the arrow in some node of Fig. 3  and it is unclear if it was used or not. I am not sure in right identification of the genus (Trichodes) because it is perhaps based on old doubtful description. No Trichodes sp. are distributed in South America now and Gondwanan distribution of Trichodes is improbable in Cretaceous. Old entomopalaeontologists often described their fossils as 'Necrobia', 'Clerus' or 'Trichodes' and I suppose it is one of them. If I am wrong and authors are sure in their generic determination (description is concise or they had studied the fossil), the fact should be highlighted in the relevant chapter because it would be also important outcome of the communication.

Reviewer 2 Report

Dear Authors,

I read  carefully your submitted mns Insects-1547905. I consider it an original and quite interesting contribution to the phylogenetic studies of the important predatory group of Coleoptera Cleridae. The methods approach is modern and shows an high scientific  "sound" by using Illumia sequencing. The M& M have been described well and clear, so as for the Results . The only missing is the section "Discussion".Moreover you provided , here and there as in M & M and Results,  several important comments that more  likely could be  moved into a Discussion chapter, separated from the other parts of the mns. The Discussion is too important for your contribution in order to improve its  scientific value. So, I attached the word version with my few suggestions for the re-drafting structure of the text....and I strongly suggest to follow  them. The references section provided is fully complete and appropriate.

Sincerely

Reviewer 3 Report

This is well written. I think OTUs are limit in Asian genera/species and the result and discussion are also limit, but acceptable for public. I pointed out the following 2 minor suggestions

1. Important finding is the sister group relationship with Clerinae and Hydnocerinae, but this was also reported by the following article. Please refer this paper and rewrite the text. 

J.S. Bartlett (2021) A Preliminary Suprageneric Classification for Clerinae (Coleoptera: Cleridae) based on Molecular and Morphological Evidence, Including a Review of Tegminal Terminology. Annales Zoologici, 71 (4): 737-766.

2. In Fig. 2, Melyridae should be not italic.

Round 2

Reviewer 1 Report

I found the article very interesting and fully agree with the author's revision and recommend for publication in the present form. Some minor suggestions in stylistics/grammer are included and some typos highlighted in the enclosed pdf.

Jiri Kolibac
